# Influenza A virus reassortment in mammals gives rise to genetically distinct within-host subpopulations

Ketaki Ganti[1], Anish Bagga[2], Silvia Carnaccini[3], Lucas M. Ferreri[1,3], Ginger Geiger[3], C. Joaquin Caceres[3], Brittany Seibert[3], Yonghai Li[4], Liping Wang[5], Taeyong Kwon[4], Yuhao Li[4], Igor Morozov[4], Wenjun Ma[5,6], Juergen A. Richt[4,6], Daniel R. Perez[3,7], Katia Koelle[8,9] & Anice C. Lowen[1,9] ✉

Influenza A virus (IAV) genetic exchange through reassortment has the potential to accelerate viral evolution and has played a critical role in the generation of multiple pandemic strains. For reassortment to occur, distinct viruses must co-infect the same cell. The spatio-temporal dynamics of viral dissemination within an infected host therefore define opportunity for reassortment. Here, we used wild type and synonymously barcoded variant viruses of a pandemic H1N1 strain to examine the within-host viral dynamics that govern reassortment in guinea pigs, ferrets and swine. The first two species are well-established models of human influenza, while swine are a natural host and a frequent conduit for cross-species transmission and reassortment. Our results show reassortment to be pervasive in all three hosts but less frequent in swine than in ferrets and guinea pigs. In ferrets, tissue-specific differences in the opportunity for reassortment are also evident, with more reassortants detected in the nasal tract than the lower respiratory tract. While temporal trends in viral diversity are limited, spatial patterns are clear, with heterogeneity in the viral genotypes detected at distinct anatomical sites revealing extensive compartmentalization of reassortment and replication. Our data indicate that the dynamics of viral replication in mammals allow diversification through reassortment but that the spatial compartmentalization of variants likely shapes their evolution and onward transmission.

Influenza A viruses (IAVs) have a broad host range, with the greatest diversity in wild bird species and established lineages circulating in poultry, swine, humans, and other mammalian hosts[1,2]. Host range for a given lineage is restricted by species barriers to infection, but occasional spillover events can result in sustained transmission, seeding new lineages[3,4]. The establishment of novel IAVs in humans is the source of influenza pandemics and has major ramifications for public health and the economy[5,6]. Reassortment of gene segments between IAVs adapted to distinct host species can give rise to chimeric viruses with enhanced potential for cross-species transfer[7]. Prominent examples include the reassortant strains that gave rise to 1957, 1968, and 2009 influenza pandemics[8,9].

Among non-human mammalian species, swine are of particular interest. Transmission of IAV between swine and humans is relatively common, due not only to an extensive interface in agricultural settings, but also to similarities between pigs and

humans in the host factors that support viral replication[10,11]. Swine hosts are also somewhat susceptible to infection with avian IAV, creating the opportunity for co-infection and reassortment of viruses that are typically separated by host species barriers[12]. For this reason, swine are often referred to as mixing vessels for IAV reassortment[13,14]. The role of swine in the 2009 influenza pandemic[9], and the abundant reassortment that characterizes IAV in swine[15–19], offer ample support for this designation. Although the central position of swine in IAV ecology and evolution is well understood[10,13,16], the within-host viral dynamics that govern reassortment in this host are less well-characterized.

We have previously reported that IAV reassortment occurs frequently in vivo[20–24]. This prior work focused on easily sampled IAV populations replicating in the upper respiratory tracts of mammals. Recent work has shown, however, that distinct within-host subpopulations can form in different regions of the respiratory tract[25]. This spatial structure, in turn, can influence viral evolutionary dynamics within and between hosts[26,27]. Here, we evaluated how genotypic diversity generated through reassortment is shaped by and contributes to viral population structure within the host.

We examined within-host diversity generated through reassortment in guinea pigs, ferrets, and pigs. While pigs are an important natural host, guinea pigs and ferrets are well-characterized models for human IAV[28–30]. We find that co-infection at the cellular level is extensive and reassortant viruses are consistently observed. In ferrets, markedly greater viral genotypic diversity is generated in the upper respiratory tract compared to the lower tract. Viral populations at these two sites are furthermore highly dissimilar genetically, revealing strong spatial compartmentalization. Diversity generated through reassortment in swine is generally lower than in guinea pigs and ferrets and, as in ferrets, viral diversity in swine is characterized by extensive spatial compartmentalization. Taken together, our data reveal spatial structure within the mammalian respiratory tract to be a major factor defining the extent of IAV diversity engendered through reassortment.

## Results

### Viral genotypic diversity generated through reassortment in the mammalian nasal tract

To evaluate the viral genotypic diversity generated via reassortment, guinea pigs, ferrets and pigs were co-infected with well-matched parental viruses of the influenza A/Netherlands/602/2009 (NL09; pH1N1) background[31]. Termed wild type (WT) and variant (VAR), these co-infecting viruses differ by a single synonymous mutation introduced into each gene segment of the VAR virus and distinct epitope tags encoded in the hemagglutinin open reading frame of each virus. In cell culture, these two viruses show comparable fitness (Supplementary Fig. 1). Following the determination of the 50% infectious dose (ID$_{50}$) of the virus mixture in ferrets and utilizing prior ID$_{50}$ results in guinea pigs[32], reassortment in these species was analyzed at two doses applied intranasally ($1 \times 10^2$ and $1 \times 10^5$ ID$_{50}$). In pigs, a single dose of $2 \times 10^6$ PFU applied both intranasally and intratracheally was used after preliminary tests with lower, intranasal, doses showed that animals were not consistently infected. All inoculated animals supported robust viral replication, as assessed at the nasal site, with titers declining to the limit of detection by days 4–7 post-inoculation (Supplementary Fig. 2).

To monitor reassortment in the upper respiratory tract, nasal samples were collected longitudinally, and the genotypes of clonal isolates derived from each sample were determined. The parental origin of each of the eight segments was defined, allowing the identification of all 254 possible reassortant genotypes and both parental genotypes. Here, a genotype is defined as a constellation of eight gene segments, and any nucleotide diversity present is not considered. Reassortment was detected in all samples from all species tested (Supplementary Fig. 3). To characterize the viral diversity

generated through reassortment, we calculated the frequency of unique genotypes, frequency of parental genotypes, richness, and Shannon–Weiner diversity for each sample (Fig. 1).

In all three host species, the same reassortant virus is rarely detected across multiple time points (Supplementary Fig. 3), suggesting genotypic loss due to stochastic processes such as drift or disruption through further reassortment. Parental genotypes (WT or VAR), however, are consistently detected in the nasal samples of each animal in least two time points (Fig. 1B, F, J, M). Despite efforts to inoculate with a 1:1 mixture of WT and VAR viruses, VAR was predominant; this could reflect a skewing in the inoculum or fitness advantage of VAR that is only apparent in vivo. Since 21 viral isolates were genotyped from each sample, richness has a maximum value of 21 in our dataset. Richness is often high in guinea pigs and ferrets, with values ranging from 4 to 18, while the range in pigs falls between 3 and 10 (Fig. 1C, G, K, N). Trends in diversity correspond to those of richness (Fig. 1D, H, L, O).

In guinea pigs and ferrets, only marginal effects of dose were apparent (Fig. 1, compare dashed and solid lines). Nevertheless, as shown in Supplementary Fig. 4, differences in diversity across doses were significant in both species, with the higher dose yielding higher diversity when all time points were considered together ($p = 0.04$ in guinea pigs and $p = 0.003$ in ferrets, ANOVA). Richness was also significantly higher in ferrets receiving a higher dose ($p = 0.002$, ANOVA).

The effects of dose are relevant in interpreting comparisons across species as only a relatively high dose was tested in pigs. Comparison of species across all time points and doses shows significantly higher richness and diversity in guinea pigs and ferrets than in pigs (Fig. 1N, O; $p < 0.01$, ANOVA). Limiting this analysis to include only animals receiving a high dose of inoculum confirmed these trends ($p < 0.005$, ANOVA).

Overall, these results reveal that the mammalian upper respiratory tract presents an environment with high potential for the generation of viral diversity via reassortment, with modest effects of dose observed and with swine showing lower viral diversity than guinea pigs or ferrets.

### Viral diversity generated through reassortment is greater in the ferret nasal tract than in the lungs

In mammalian hosts, IAV replication can be distributed throughout the respiratory tract. Using the ferret model, we compared the extent of viral diversity arising through reassortment in upper and lower respiratory tracts. Nasal turbinates and lung tissues were collected on days 1, 2, 3, or 4 from ferrets inoculated with $1 \times 10^2$ or $1 \times 10^5$ ferret ID$_{50}$. Viral titers in these samples are reported in Supplementary Fig. 2. Genotyping of viral isolates revealed that the number of unique genotypes detected was routinely higher in nasal turbinates compared to lung at both doses (Fig. 2A–C). Similarly, Shannon–Weiner diversity in nasal turbinates was typically higher than in the lung (Fig. 2D). These differences in both richness and diversity were significant (Fig. 2E, F) ($p = 8.6 \times 10^{-5}$ and $p = 0.0014$, respectively; paired $t$ test).

We hypothesized that the distinct outcomes of co-infection in the upper and lower respiratory tracts were indicative of little viral dispersal between the two sites. To test this hypothesis, we modeled free mixing between the lung and nose by computationally shuffling intact genotypes between the two tissues. We then compared our observed data to the resultant simulated dataset (Fig. 2C, D). This approach revealed that richness and diversity seen in the lungs are low relative to that expected for free mixing, with observed values typically falling below the 5th percentile of simulated results. In contrast, richness and diversity observed in nasal turbinates are typically above the mean. This analysis suggests that the spread of the virus between upper and lower respiratory locations occurs relatively rarely, such that abundant diversity generated through reassortment in the nasal tract does not seed a similarly diverse population in the lungs.

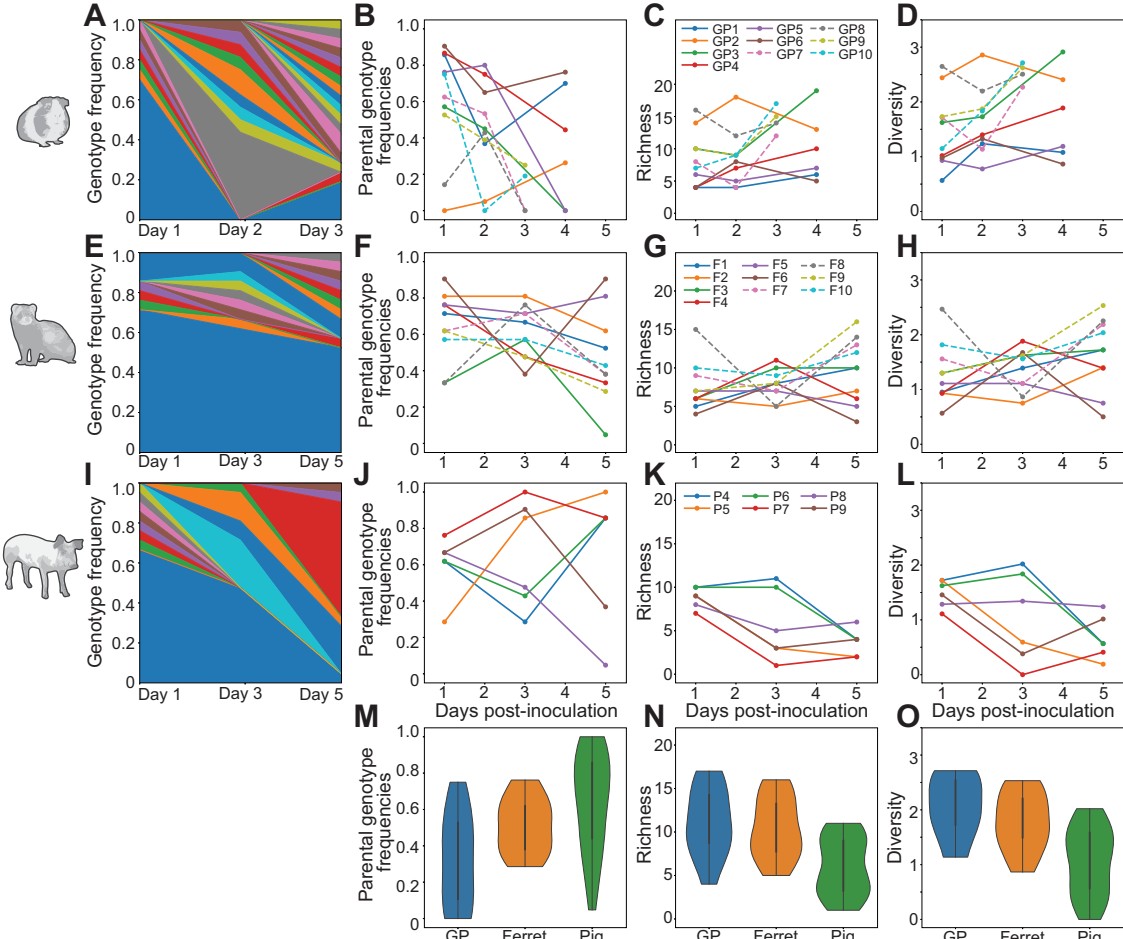

**Fig. 1 | Viral genotypic diversity generated through reassortment in the mammalian nasal tract.** Results from guinea pigs are shown in **A**–**D**, ferrets in **E**–**H**, and swine in **I**–**L**. Stacked plots (**A**, **E**, **I**) show frequencies of unique genotypes detected in one representative animal over time. Blue and orange represent the WT and VAR parental genotypes, respectively. The frequency of both parental genotypes combined (**B**, **F**, **J**), richness (**C**, **G**, **K**), and Shannon–Weiner diversity (**D**, **H**, **L**) are each plotted as a function of time. Guinea pigs and ferrets inoculated at high dose ($1 \times 10^5$ ID$_{50}$) and low dose ($1 \times 10^2$ ID$_{50}$) are indicated with dashed and solid lines, respectively. The distribution of parental genotype frequencies (**M**), richness (**N**), and diversity (**O**) across all time points in each species (guinea pigs $n = 12$; ferrets $n = 12$; swine $n = 18$) is shown with violin plots. Data are presented as violin plots featuring a box plot. The bounds of the box show the first and third quartiles. Whiskers indicate 1.5 times the interquartile range and contain ~99% of the data for a normal distribution. The bounds of the violin plots indicate the minima and maxima of the entire dataset. Differences in richness and diversity between swine and guinea pigs (richness $p = 0.00065$; diversity $p = 0.00014$) and between swine and ferrets (richness $p = 0.0016$; diversity $p = 0.0028$) were significant. Parental genotype frequencies differed significantly between swine and guinea pigs ($p = 0.003$) and were not significant between swine and ferrets ($p > 0.05$). All statistics were derived using one-way ANOVA. Animal silhouettes were generated using BioRender.com.

## Reassortant viral populations in the ferret upper and lower respiratory tracts are distinct

Examination of the specific genotypes identified in the nasal turbinates and lungs within a given ferret revealed almost no overlap (Fig. 2A, B). To more rigorously assess this apparent dissimilarity of viral populations, we calculated normalized beta diversity ($\beta_n$) between pairs of samples[33]. This dissimilarity metric has a range between 0 and 1, with a beta value of 0 indicating that two samples are identical in their genotype composition, and a beta value of 1 indicating no genotypic overlap. In all ferrets, high beta diversities of >0.5 are seen between nasal turbinate and lung samples of the same animal (Fig. 3A). Indeed, beta diversity values calculated between pairs of samples within and between ferrets indicate that the nasal and lung viral populations within an individual are typically as distinct as populations replicating in different animals (Fig. 3A). To test whether a lack of viral dispersal between anatomical sites could account for this observation, we compared the observed beta diversity to the distribution of beta diversity values obtained from the computational simulation of free mixing outlined above. For most ferrets, observed beta diversity falls above the 95th percentile of the distribution (Fig. 3B). Thus, the highly distinct viral populations seen in nasal and lung tissues are inconsistent with free mixing between these sites. These data further support the notion that virus replicating within the ferret upper and lower respiratory tracts form distinct compartments, with reassortment occurring independently at the two sites.

To visualize the spatial distribution of co-infecting viruses within ferret tissues, the tagged HA proteins of WT and VAR origin were stained within lung and nasal tissue sections (Fig. 3C). This analysis revealed that microscopic fields that are positive for viral antigen tend to show a high density of infected cells at both tissue sites. Strikingly, most infected cells are positive for both WT and VAR antigens, indicating high levels of viral co-infection within the host, consistent with the robust reassortment observed. The cell types hosting viral antigen in the infected lung tissue were defined based on their morphology and found to comprise ciliated epithelium of the bronchi and bronchioles, goblet cells, cells of the peribronchial/-iolar submucosal glands, and type II pneumocytes within the alveoli (Fig. 3D and Supplementary Data 2). Unfortunately, this

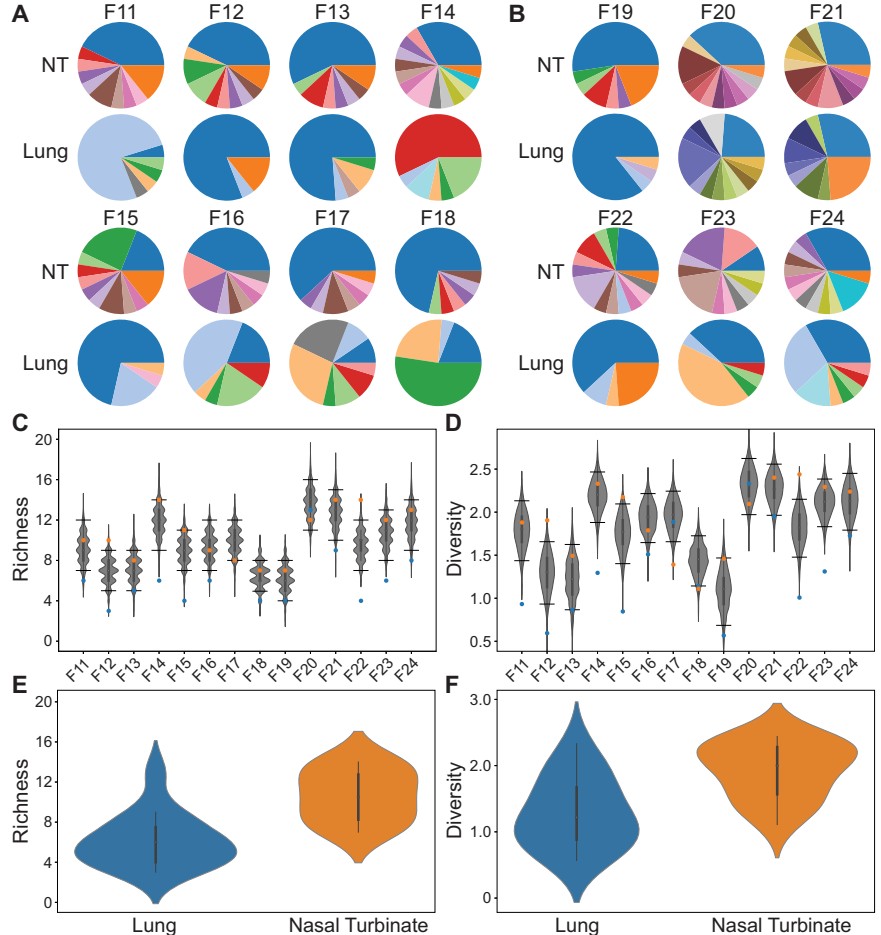

**Fig. 2 | Viral diversity generated through reassortment is greater in the ferret nasal turbinates than in the lungs.** Pie charts show frequencies of unique genotypes detected in the nasal turbinates (NT) and lung of ferrets inoculated with $1 \times 10^2$ FID$_{50}$ (**A**) or $1 \times 10^5$ FID$_{50}$ (**B**). Ferrets shown were analyzed at the following times post-infection: Day 1 (F11, 12, 19, 20), Day 2 (F13, 14, 21, 22), Day 3 (F15, 16, 23), and Day 4 (F17, 18, 24). Blue and orange sections of the pie charts represent VAR and WT parental genotypes, respectively. Richness (**C**) and diversity (**D**) are plotted, with observed results (colored points) overlaid on the distribution of simulated data (gray violins) (*n* = 1000 simulations per ferret). Horizontal lines denote the 95th and 5th percentiles. The distribution of richness (**E**) and diversity (**F**) in NT and lungs of all individuals are shown with violin plots (Lung *n* = 14; Nasal turbinates *n* = 14). Data are presented as violin plots featuring a box plot. Data are presented as violin plots featuring a box plot. The bounds of the box show the interquartile range and the center of the box shows the 50th percentile. The median is shown by the white dot. Whiskers indicate 1.5 times the interquartile range and contain ∼99% of the data for a normal distribution. The bounds of the violin plots indicate the minima and maxima of the entire dataset. Differences between NT and lungs were significant by both measures (*p* < 0.01, two-sided paired *t* test).

analysis could not be performed for nasal turbinate owing to the exhaustion of the samples.

## Local diversity arising through reassortment is modest within the swine respiratory tract

Analysis of nasal swabs showed limited genotypic diversity in the swine nasal tract. We expanded this investigation of reassortment in pigs by sampling the nasal tract and each of the seven lung lobes of groups of three pigs on day 3 or 5 post-inoculation. Viral titers in these samples are shown in Supplementary Fig. 2. Titers in the nasal tract on day 3 were insufficient to support analysis of reassortment. Genotype frequencies observed from the remaining sites were plotted and overlaid on a schematic of a pig lung (Fig. 4A–F). Although one or more reassortants are detected in all but three of the 45 samples, the viral populations tend to carry high proportions of parental virus. At a minority of sites, a specific reassortant is predominant. Richness is less than 10 for most samples (Fig. 4G) and Shannon–Weiner diversities less than 1.0 are common (Fig. 4H). Thus, overall, the diversity generated through reassortment throughout the swine respiratory tract is moderate. To evaluate the extent to which a lack of viral mixing across the sampled locations limits richness and diversity, we compared the

observed results to those expected if viruses mix freely throughout the respiratory tract (Fig. 4G, H). To simulate free mixing, we randomly sampled intact viral genotypes from all those detected within a given pig. The resultant richness and Shannon–Weiner diversity values show wide distributions. However, much of the observed data lies below the 5th percentile of these distributions, suggesting that pockets of low richness and diversity persist locally due to an absence of mixing across anatomical sites. In sum, local genotypic diversity generated via reassortment is modest throughout the swine respiratory tract.

## Reassortant viral populations show extensive compartmentalization within the swine respiratory tract

Examination of the reassortant viral genotypes identified in swine tissues reveals little to no overlap across locations within an individual animal (Fig. 4A–F). We again used beta diversity to assess the similarity of viral populations in a pairwise and quantitative manner. As seen in ferrets, beta values in pigs were typically high (Fig. 5A and Supplementary Fig. 5). Indeed, viral populations sampled from within a pig were as distinct from one another as those sampled from different pigs (Fig. 5A). To test whether a lack of viral dispersal between anatomical sites could account for this observation, we compared the beta

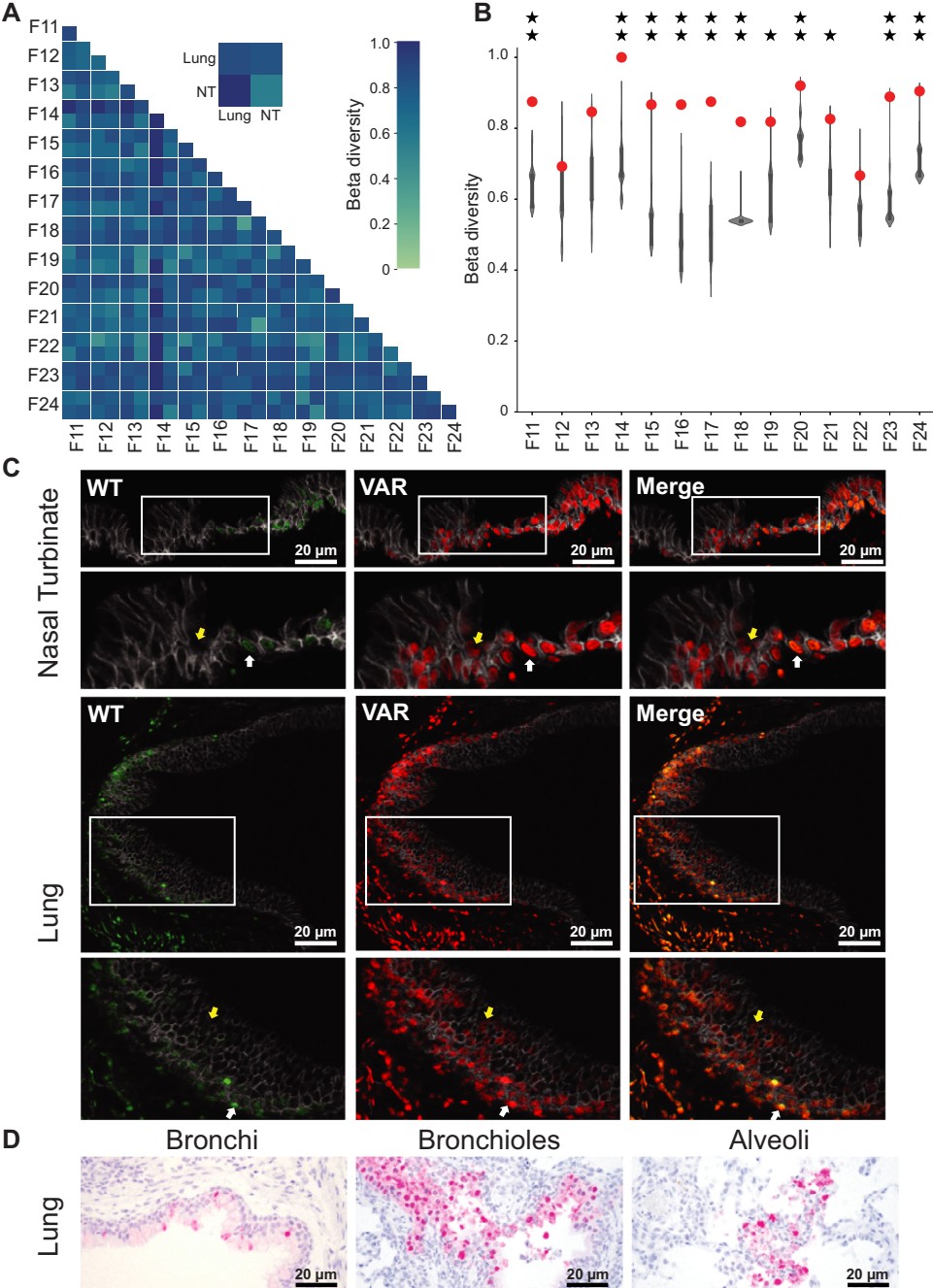

**Fig. 3 | Reassortant viral populations in the ferret upper and lower respiratory tracts are distinct.** Heat map showing normalized beta diversity of viral populations in ferret lung and nasal turbinates (NT) (**A**). The inset shows a representative comparison between the tissues of ferrets F23 and F21, to indicate the position of NT and lung within the matrix. Normalized beta diversity is plotted in **B** with observed results (colored points) overlaid on the distribution of simulated data (gray violins) (*n* = 1000 simulations per ferret). Data are presented as violin plots featuring a box plot. Data are presented as violin plots featuring a box plot. The bounds of the box show the interquartile range and the center of the box shows the 50th percentile. Whiskers indicate 1.5 times the interquartile range and contain ~99% of the data for a normal distribution. The bounds of the violin plots indicate the minima and maxima of the entire dataset. One star indicates that observed data is above the 95th percentile of the distribution; two stars indicates that observed data is above the 99th percentile. **C** Immunohistochemistry images of ferret F23 NT and lung sections stained for WT (green) and VAR (red) viruses at day 3 post inoculation. Gray staining marks epithelial cell borders. Yellow coloring in merged images indicates the presence of both WT and VAR HA antigens in the same cell. Zoomed insets of both NT and lung sections are shown with white arrows indicating co-infected cells and yellow arrows indicating singly infected cells. Scale bars are 20 μm. Four fields were analyzed for each tissue section and representative images are depicted here. **D** Immunohistochemistry images of ferret F24 lung sections stained for nucleoprotein (red) and counterstained with hematoxylin. Scale bars are 20 μm. Four fields were analyzed for each tissue section and representative images are depicted here.

diversity observed for a pair of locations within a pig to the distribution of beta diversity values obtained from a simulation of free mixing between those locations (Fig. 5B and Supplementary Fig. 6). For most pairs of tissues, the observed data are at the high extreme of this distribution, falling within the 95th percentile of the simulated results.

Thus, spatial compartmentalization within the swine respiratory tract is extensive.

To visualize the spatial distribution of co-infecting viruses within swine tissues, the tagged HA proteins of WT and VAR origin were stained (Fig. 5C and Supplementary Fig. 7). As in ferrets, a high density

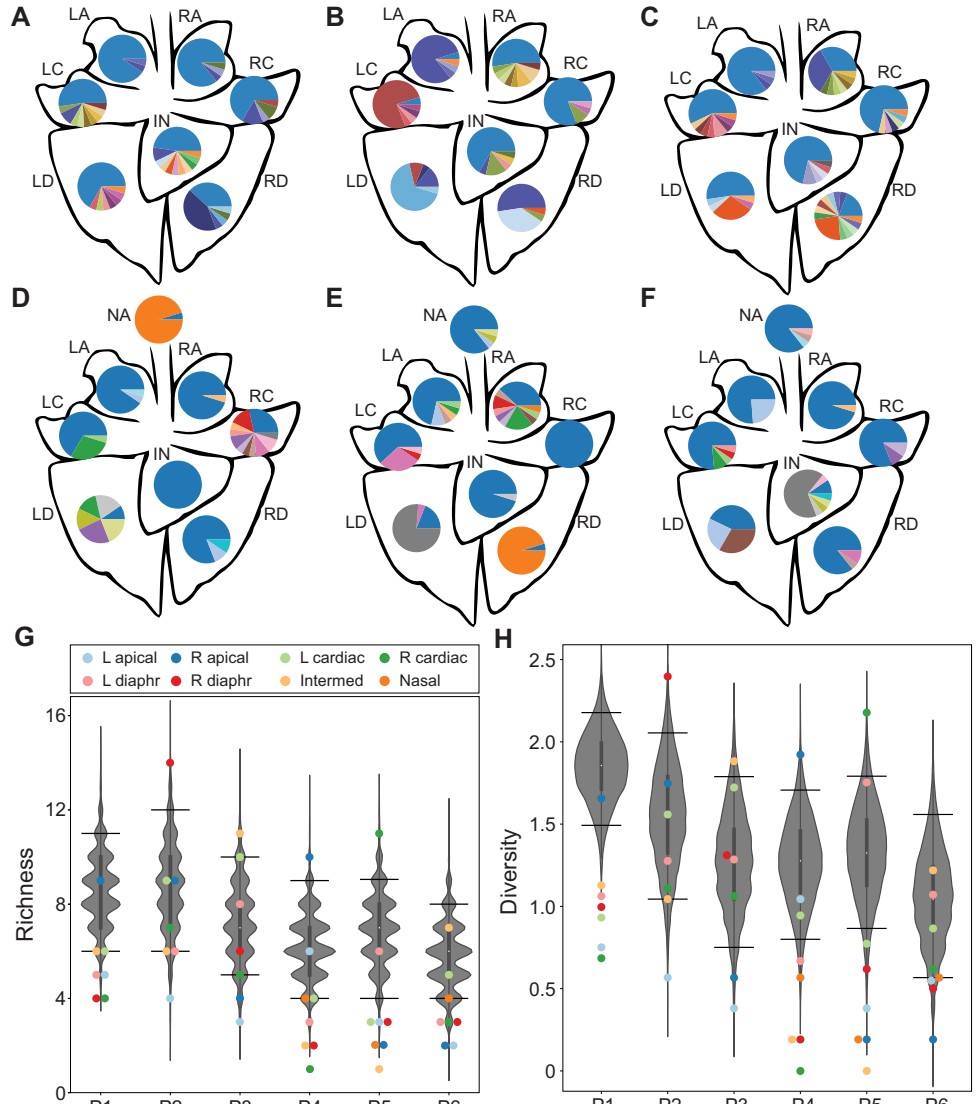

**Fig. 4 | Local diversity arising through reassortment is modest within the swine respiratory tract.** Pie charts showing frequencies of unique genotypes present within each lung lobe of a given pig on day 3 (**A**–**C**) and the nasal tract and each lung lobe of a given pig on day 5 (**D**–**F**) with blue and orange sections representing WT and VAR parental genotypes, respectively. The tissue sites are abbreviated as NA-Nasal; LA-Left Apical; RA-Right Apical; LC-Left Cardiac; RC-Right Cardiac; LD-Left Diaphragmatic; RD-Right Diaphragmatic; IN-Intermediate (accessory). Simulated richness (**G**) and diversity (**H**) are plotted, with observed results (colored points) overlaid on the distribution of simulated data (gray violins) ($n = 1000$ simulations per pig). Data are presented as violin plots featuring a box plot. The bounds of the box show the interquartile range and the center of the box shows the 50th percentile. The median is shown by the white dot. Whiskers indicate 1.5 times the interquartile range and contain ∼99% of the data for a normal distribution. The bounds of the violin plots indicate the minima and maxima of the entire dataset. Horizontal lines denote the 95th and 5th percentiles.

of infected cells is visible in regions positive for viral antigen and most infected cells are positive for both WT and VAR viruses. Thus, the high levels of viral co-infection observed in ferrets extends to swine. Using a morphologic approach to identify cell types, we found type II pneumocytes within the alveoli and ciliated epithelial cells of the bronchioles and bronchi to be the predominant cell types harboring viral antigen in the lungs (Fig. 5D, Supplementary Fig. 8 and Supplementary Data 2). In nasal turbinates, viral antigen positivity was observed within respiratory and transitional epithelia (Supplementary Fig. 8 and Supplementary Data 2).

## Discussion

Our experiments reveal the extent to which co-infection at the level of the whole host creates opportunity for reassortment within the host. Abundant opportunity is apparent in the nasal tracts of guinea pigs and ferrets. However, the viral diversity generated through reassortment is more limited in the lungs of ferrets and all tissues sampled in pigs. In addition to relatively little opportunity for reassortment in these tissues, spatial constraints on mixing within the respiratory tract yield localized subpopulations with diminished genotypic diversity. Thus, despite the extensive reassortment seen among swine IAV at a population level, our results reveal only moderate viral diversity generated through reassortment in pigs. Results from guinea pigs and ferrets suggest that opportunity for within-host IAV genetic exchange may be greater in other natural mammalian hosts.

In drawing comparisons across the species examined, it is important to consider differences in the protocols applied. Swine were infected both intranasally and intratracheally compared to only intranasal infection in both ferrets and guinea pigs. The delivery of inoculum directly to the lower respiratory tract in swine might be expected to augment potential for WT-VAR co-infection in the lungs. In addition, swine were infected with a single high dose with unknown relationship

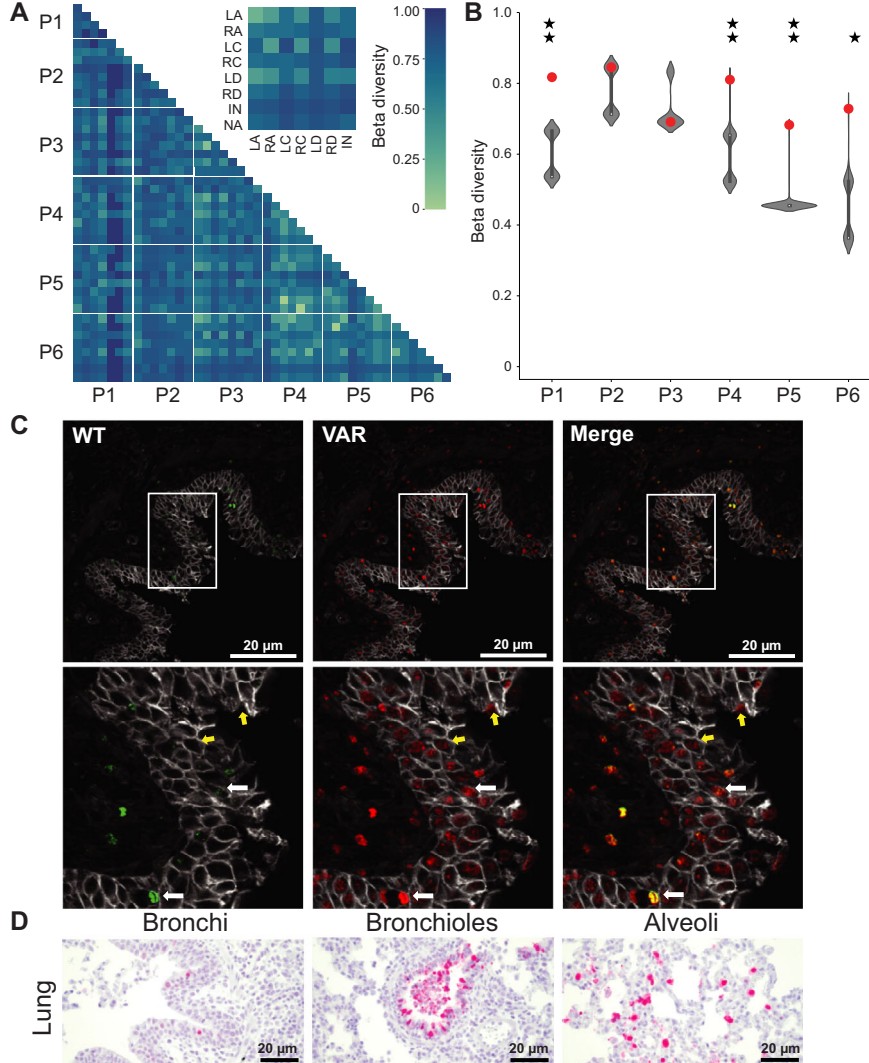

**Fig. 5 | Reassortant viral populations show extensive compartmentalization within the swine respiratory tract.** Heat map showing normalized beta diversity of viral populations in swine lung and nasal tract (**A**). The inset shows a representative comparison between the tissues of pigs P3 and P6, to indicate the position of each tissue within the matrix. The tissue sites are abbreviated as NA-Nasal; LA-Left Apical; RA- Right Apical; LC-Left Cardiac; RC-Right Cardiac; LD-Left Diaphragmatic; RD-Right Diaphragmatic; IN-Intermediate (accessory). Normalized beta diversity between the left apical and left cardiac lobes is plotted (**B**) with observed results (colored points) overlaid on the distribution of simulated data (gray violins) (*n* = 1000 simulations per pig). Data are presented as violin plots featuring a box plot. Data are presented as violin plots featuring a box plot. The bounds of the box show the interquartile range and the center of the box shows the 50th percentile. Whiskers indicate 1.5 times the interquartile range and contain ~99% of the data for a normal distribution. The bounds of the violin plots indicate the minima and maxima of the entire dataset. One star indicates that observed data is above the 95th percentile of the distribution; two stars indicates that observed data is above the 99th percentile. **C** Representative immunohistochemistry images of swine (P76) left apical lung section stained for WT (green) and VAR (red) viruses at day 3 post-inoculation. Gray staining marks epithelial cell borders. Yellow coloring in merged images indicates the presence of both WT and VAR HA antigens in the same cell. Zoomed insets of the lung are shown with white arrows indicating co-infected cells and yellow arrows indicating singly infected cells. Scale bars are 20 μm. Four fields were analyzed for each tissue section and representative images are depicted here. **D** Immunohistochemistry images of swine (P76) left apical lung sections stained for nucleoprotein (red) and counterstained with hematoxylin. Scale bars are 20 μm. Four fields were analyzed for each tissue section and representative images are depicted here.

to the median infectious doses in this species, while ferrets and guinea pigs were each infected with doses of $1 \times 10^2$ and $1 \times 10^5$ ID$_{50}$. As measured by plaque assay, the dose applied in pigs is comparable to the high dose used in guinea pigs and approximately 10-fold lower than the high dose used in ferrets. While inoculum dose might be expected to influence the potential for co-infection and propagation of novel variants, the effect of a $10^3$-fold difference in dose in ferrets and guinea pigs were modest. We, therefore, suggest that the more limited diversification through reassortment in pigs is unlikely to be due to protocol differences.

Given our use of well-matched parental viruses, the differences in viral genotypic diversity seen between species and between tissues within a species are also unlikely to be driven by selection. Instead, features of viral dynamics in these different environments likely shape the frequency of reassortment and the extent to which reassortants are amplified once formed. Since relatively few reassortants were detected in pigs and in ferret lungs, our data imply that the size of the target tissue may be an important factor. In a large animal such as swine or tissue with extensive surface area, such as lung, the likelihood of cellular co-infection involving genetically distinct variants may be reduced. In addition, viral fitness, the spatial arrangement of target cells, the extent to which permissive cells are interspersed with non-permissive cells, physical barriers to viral spread, and the effectiveness innate responses to

infection are all likely to vary with species and across different tissues within a species[34–36].

Although variation in the diversity generated through reassortment is apparent, visualization of viral antigen in infected tissues indicated that cellular co-infection is abundant at all sites examined. Cellular co-infection has previously been reported to be common in vivo[37] and is expected owing to the high potential for singular genomes to result in abortive infection[38,39] and the potential for direct cell-to-cell spread of viral genetic material[40]. Nevertheless, the juxtaposition of this result with a predominance of parental-type viruses in many samples is counter-intuitive. While the images of fixed tissue show a snapshot of viral HA protein expression in defined cells at a given time point, viral samples draw from the population of viruses that have accumulated at a particular site over the course of infection. Thus, the prevalence of parental genotypes may reflect their production early after inoculation, when cellular co-infection is less likely to occur[22,37]. In addition, HA staining does not reveal the relative dosage of WT and VAR genomes in a cell or indicate the genotype(s) of the seven non-HA segments. Both factors would strongly modulate the proportion of progeny genomes from a co-infected cell that is reassortant.

The spatial separation of viral subpopulations within an individual has important evolutionary implications. First, reassortment among genetically distinct viruses is less likely to occur when viral dispersal is localized, since all viruses produced from a singly infected progenitor cell will be genetically similar[23]. Conversely, well-mixed viral populations are more likely to produce reassortants that differ from their parents in biologically meaningful ways. Compartmentalization of variants may be a contributing factor to the lack of reassortment reported in experimentally infected humans[41]. Second, spatial structure impacts the extent to which a viral population within a host is shaped by selection versus stochastic processes[25,42]. Spatial barriers create subpopulations, each of which can have a small population size. As such, genetic drift dominates within each subpopulation, with variant frequencies changing over time largely due to chance. This means that both lower and higher fitness variants have the potential to rise to high frequencies within subpopulations. As a result, spatial compartmentalization of populations has the immediate effect of reducing the efficiency of selection. While the strength of selection to purge low-fitness variants is weakened at the level of individual subpopulations, the propagation of slightly deleterious mutants allows a broader exploration of sequence space across the entire viral population that is distributed within the host. This may, in rare cases, lead to the discovery of higher fitness variants that might not evolve as readily in less structured within-host populations.

Compartmentalization of viral replication is also expected to shape evolution occurring between hosts. As has been seen in ferrets, transmitted virus is likely to comprise a sub-sample of one specific anatomical location[23,27,43]. In the context of spatial heterogeneity, this sub-sample will not be representative of the viral population present in the whole host. Thus, within-host spatial structure is expected to contribute to stochastic effects acting between hosts: the transmitted virus will be the virus that was in the right place at the right time, and may not be the most fit genotype present within a host[23,27,44,45].

Certain limitations in our experimental design are important to consider. Our approach of co-inoculating with well-matched parental strains is designed to maximize potential to detect reassortment. In addition, the gene constellation of the 2009 pandemic H1N1 virus, used in this study, has shown a marked predisposition for reassortment[46–48]. However, natural co-infections will include a wide range of scenarios in terms of relative timing, route and dose of infection, and phenotypes of co-infecting strains. Each of these factors will influence the frequency of reassortment[21–23,49]. In evaluating the extent of overlap between viral genotypes in different anatomical locations, it is important to note that parental viruses are likely seeded

throughout the respiratory tract upon inoculation; thus, only reassortant genotypes are informative in assessing the extent of mixing across sites. In addition, if variants are rare (<5%), they would fall below the limit of detection of our assay, which can have potent effects on measured beta diversity. In particular, if a genotype detected in one location is present below the limit of detection in a second location, measured beta diversity would not reflect this commonality and would therefore be erroneously high.

IAV reassortment within cells is efficient, such that 256 distinct genotypes are readily formed in a single co-infected cell[50]. However, within a host, the high potential for reassortment to generate diversity is subject to complex dynamics that define the likelihood of cellular co-infection and the extent to which novel reassortants are propagated. While our data reveal extensive co-infection in vivo and consistent formation of reassortants, species and tissue differences in the extent of reassortment are apparent. Spatial constraints on the dissemination of novel genotypes add to this complexity and are likely to be a major factor in within-host viral evolutionary dynamics.

## Methods

### Ethical considerations
All the animal experiments were conducted in accordance with the Guide for the Care and Use of Laboratory Animals of the National Institutes of Health. The studies were conducted under animal biosafety level 2 containment and approved by the IACUC of Emory University (DAR-2002738-ELMNTS-A) for guinea pig (*Cavia porcellus*), the IACUC of the University of Georgia (AUP A2015 06-026-Y3-A5) for ferret (*Mustela putorius furo*) and the IACUC of Kansas State University (protocol #4120) for swine (*Sus scrofa*). The animals were humanely euthanized following guidelines approved by the American Veterinary Medical Association.

### Cells and cell culture media
Madin–Darby canine kidney (MDCK) cells, a gift from Dr. Robert Webster, St Jude Children's Research Hospital, Memphis, TN to D.R.P were used for all experiments. A seed stock of MDCK cells at passage 23 was subsequently amplified and maintained in Minimal Essential Medium (Gibco) supplemented with 10% fetal bovine serum (FBS; Atlanta Biologicals) and Normocin (Invivogen). 293 T cells (ATCC, CRL-3216) were maintained in Dulbecco's Minimal Essential Medium (Gibco) supplemented with 10% FBS and PS. All cells were cultured at 37 °C and 5% $CO_2$ in a humidified incubator. The cell lines were not authenticated. All cell lines were tested monthly for mycoplasma contamination while in use. The medium for the culture of IAV in MDCK cells (virus medium) was prepared by supplementing the basal medium for the relevant cell type with 4.3% BSA and Normocin.

### Viruses
Viruses used in this study were derived from influenza A/Netherlands/602/2009 (H1N1) virus (NL09) and were generated by reverse genetics[51–53]. In brief, 293 T cells transfected with reverse genetics plasmids 16–24 h previously were co-cultured with MDCK cells at 37 °C for 40–48 h. Recovered virus was propagated in MDCK cells at a low multiplicity of infection to generate working stocks. Titration of stocks and experimental samples was carried out by plaque assay in MDCK cells. Silent mutations were introduced into each segment of the VAR virus by site-directed mutagenesis of reverse genetics plasmids. The specific changes introduced into the VAR virus were reported previously[24,31]. NL09 VAR virus was engineered to contain a 6XHIS epitope tag plus a GGGS linker at the amino (N) terminus of the HA protein following the signal peptide. NL09 WT virus carries an HA epitope tag plus a GGGS linker inserted at the N terminus of the HA protein[31]. For animal challenges, 1:1 mixture of NL09 WT and VAR viruses was prepared using methods described previously[24]. This mixture was validated in cell culture by quantifying cells positive for

HIS and HA tags following infection of MDCK cells, revealing an empirically determined ratio of 0.95:1 (WT:VAR). The same mixture was used for all experiments reported herein.

## Evaluation of viral replication in cell culture

Replication of NL09, NL09 WT, and NL09 VAR viruses was determined in triplicate culture wells. MDCK cells in 6 well dishes were inoculated at an MOI of 0.05 PFU/cell in PBS. After 1 h incubation at 37 °C, inoculum was removed, cells were washed 3x with PBS, 2 mL virus medium was added to cells, and dishes were returned to 37 °C. A 120 ul volume of culture medium was sampled at the indicated times points and stored at −80 °C. Viral titers were determined by plaque assay on MDCK cells.

## Animal models and reassortment in vivo

Female, Hartley strain guinea pigs weighing 250–350 g were obtained from Charles River Laboratories and housed by Emory University Department of Animal Resources. Before intranasal inoculation and nasal washing, the guinea pigs were anaesthetized with 30 mg kg$^{-1}$ ketamine and 4 mg kg$^{-1}$ xylazine by intramuscular injection. The GPID$_{50}$ of the NL09 virus was previously determined to be $1 \times 10^1$ PFU[32]. To evaluate reassortment kinetics in guinea pigs, groups of six animals were infected with $1 \times 10^3$ PFU ($1 \times 10^2$ ID$_{50}$) or $1 \times 10^6$ PFU ($1 \times 10^5$ ID$_{50}$) of the NL09 WT/VAR virus mixtures. Virus inoculum was given intranasally in a 300 µl volume of PBS. Nasal washes were performed on days 1–6 post-inoculation and titered by plaque assay. Viral genotyping was performed on samples collected on days 1, 2, and 3 or 4 for each guinea pig. Day 3 was used for animals receiving the higher dose since the virus is cleared rapidly in this system and shedding has ceased by day 4.

Female ferrets, 20-weeks-old, from Triple F Farms (Gillett, PA) were used. All ferrets were seronegative by anti-nucleoprotein (anti-NP) influenza virus enzyme-linked immunosorbent assay, Swine Influenza Virus Ab Test, (IDEXX, Westbrook, ME) prior to infection. Five days prior to experimentation, ferrets were sedated, and a subcutaneous transponder (Bio Medic Data Systems, Seaford, Delaware) was implanted to identify each animal and provide temperature readings. Anesthetics were applied via intramuscular injection with ketamine (20 mg kg$^{-1}$) and xylazine (1 mg kg$^{-1}$). Infections were performed via intranasal inoculation of 1 mL of virus diluted in PBS. Ferret nasal washes were carried out as follows. Ferrets were anesthetized and 1 ml of PBS administered to the nose was used to induce sneezing. Expelled fluid was collected into Petri dishes and samples were collected in an additional volume of 1 mL PBS. Infected ferrets were monitored daily for clinical signs, temperature, and weight loss. Ferrets were euthanized by intravenous injection of 1 ml of Beuthanasia-D diluted 1:1 with DI water (Merck, Madison, NJ).

For determination of ferret ID$_{50}$, six groups of four ferrets each were inoculated with increasing doses of the NL09 WT/VAR virus mixture ($1 \times 10^{0.1}$ PFU, $1 \times 10^0$ PFU, $1 \times 10^1$ PFU, $1 \times 10^2$ PFU, $1 \times 10^3$ PFU, and $1 \times 10^4$ PFU). Nasal washes were collected daily for up to 6 days and titrated for viral shedding by plaque assay. The ferret ID$_{50}$ was determined based on results obtained on day 2 and found to be equivalent to $3.2 \times 10^2$ PFU.

For analysis of reassortment frequency and detection of viral antigen in tissues, ferrets were inoculated with $3.2 \times 10^4$ PFU ($1 \times 10^2$ ID$_{50}$) or $3.2 \times 10^7$ PFU ($1 \times 10^5$ ID$_{50}$). After infections, nasal washes were collected daily for up to 6 days and titrated by plaque assay. Viral genotyping was performed on samples collected on days 1, 3, and 5 for each ferret. Necropsies were performed on days 1–4 for the collection of nasal turbinate and lung tissues. A single lung lobe (the left caudal lobe) was sampled from each ferret. Tissue sections collected for virology were disrupted in 1 mL of sterile PBS using the TissueLyser LT (Qiagen, Germantown, MD) at 30 Hz for 5 min twice, in microcentrifuge tubes with 3 mm Tungsten Carbide Beads (Qiagen, St. Louis,

MO). Supernatants were clarified by centrifugation and frozen at −80 °C until viral titration. For histology, tissues were submerged in 10% buffered formalin (Sigma Aldrich, St. Louis, MO) and stored at room temperature until evaluation.

The pig study was conducted at the Large Animal Research Center (a biosafety level 2+ facility) at Kansas State University in accordance with the Guide for the Care and Use of Agricultural Animals in Research and Teaching of the U.S. Department of Agriculture. To determine virus reassortment and viral antigen in tissues, 18 4-week-old influenza H1 and H3 subtype virus- and porcine reproductive and respiratory syndrome virus-seronegative gender-mixed crossbred pigs were randomly allocated into groups. Each pig was inoculated with $2 \times 10^6$ PFU of NL09 WT/VAR mixture through both intranasal and intratracheal routes ($10^6$ PFU was administered in a 1 ml volume by each of these two routes) under anesthesia as described previously[54]. Clinical signs for all experimental pigs were monitored daily throughout the experiment. Nasal swabs were collected at 1-, 3-, 5-, and 7-days post infection from each pig. Three infected pigs were euthanized at 3-, 5-, and 7-days post infection. During necropsy, nasal turbinate, trachea, and lung tissues from seven lobes collected from each pig were frozen at −80 °C for virus isolation and fixed in 10% buffered formalin for IHC examination.

## Quantification of reassortment

Reassortment frequencies were evaluated by genotyping 21 clonal viral isolates per sample as described previously[50]. This analysis was applied to guinea pig nasal washes, ferret nasal washes, swine nasal swabs, ferret tissue homogenates, and swine tissue homogenates. Time points to be examined were chosen based on positivity in all animals in a treatment group. Thus, nasal wash samples from days 1, 2, 3, or 4 were evaluated from guinea pigs while samples from days 1, 3, and 5 were evaluated for swine and ferrets. Ferret tissues collected on days 1, 2, 3, and 4 and swine tissues collected on days 3 and 5 were analyzed.

Briefly, plaque assays were performed on MDCK cells in 10 cm dishes to isolate virus clones. Serological pipettes (1 ml) were used to collect agar plugs into 160 µl PBS. Using a ZR-96 viral RNA kit (Zymo), RNA was extracted from the agar plugs and eluted in 40 µl nuclease-free water (Invitrogen). Reverse transcription was performed using Maxima reverse transcriptase (RT; ThermoFisher) according to the manufacturer's protocol. The resulting cDNA was diluted 1:4 in nuclease-free water and each cDNA was combined with segment-specific primers (Supplementary Data 3)[24,31] designed to amplify a region of approximately 100 base pairs. The amplicon for each segment contains the site of the single nucleotide change in the VAR virus. Quantitative PCR was performed with Precision Melt Supermix (Bio-Rad) using a CFX384 Touch Real-Time PCR Detection System (Bio-Rad). Quantitative PCR data was collected using CFX Manager Software v2.1 (Bio-Rad). Template amplification was followed by high-resolution melt analysis to differentiate the WT and VAR amplicons[55]. Precision Melt Analysis software v1.2 (Bio-Rad) was used to determine the parental origin of each gene segment based on the melting properties of the cDNA amplicons relative to WT and VAR controls. Each plaque was assigned a genotype based on the combination of WT and VAR genome segments, with two variants on each of eight segments allowing for 256 potential genotypes.

## Immunohistochemistry and imaging

Tissue samples from nasal turbinates of ferrets, the right caudal lung lobe of ferrets, and all seven lung lobes of swine were fixed in 10% neutral buffered formalin for at least 24 h before being embedded in paraffin. Nasal turbinates were decalcified prior to being embedded in paraffin. Sections from all the tissues were cut and slides were prepared. The tissues were deparaffinized by warming the slides at 60 °C on a slide warmer for 45 min followed by immersion in xylenes (Sigma) for 25 min. The slides were then immersed in 100% ethanol for 10 min, 95% ethanol for 10 min, and 70% ethanol for 5 min. The slides were then

washed by placing them in deionized water for 1 h. Antigen retrieval was performed by steaming the slides in 10 mM citric acid, pH 6.0 for 45 min, followed by washing in tap water and 1× PBS (Corning) for 5 min. The WT and VAR viruses were detected in the tissues using a mouse anti-HA Alexa Fluor 488 (Invitrogen catalog number A-21287; clone 16B12; 1:50 dilution) and mouse anti His Alexa Fluor 555 (Invitrogen catalog number MA1-135-A555; clone 4E3D10H2/E3; 1:50 dilution) while epithelial cell borders were stained using rabbit anti-Na$^+$K$^+$ ATPase Alexa Fluor 647 (Abcam catalog number 198367; clone EP1845Y; 1:100 dilution) at 4 °C overnight. Slides were washed three times in 1× PBS (Corning) and once in deionized water to remove excess antibody. The slides were mounted onto glass coverslips using ProLong Diamond Anti Fade mounting media (ThermoFisher). The images were acquired using an Olympus FV1000 Confocal Microscope at ×60 magnification under an oil immersion objective. The specificity of the antibodies was confirmed by infecting MDCK cells with either the NL09 WT, NL09 VAR, or both viruses for 24 h. The cells were fixed using 4% paraformaldehyde (Alfa Aesar) and stained for HA and His tags using the antibodies as described above (Supplementary Fig. 9).

For morphological analysis via IHC, the slides were pre-treated in pH 9.0 buffer at 110 °C for 15 min. Blocking was performed using hydrogen peroxide for 20 min followed by PowerBlock (BioGenex) for 5 min. Slides were washed with PBS thrice and NP antigen was detected using a goat anti-influenza NP polyclonal antibody (abcam catalog number ab155877; 1:1000 dilution) for 1 h. Slides were washed thrice with PBS to remove excess antibody and incubated with a rabbit anti-goat biotinylated IgG (Vector laboratories catalog number BA-5000; 1:5000 dilution) for 10 min. After washing, 4Plus Alkaline Phosphatase Label (BioCare Medical) was added for 10 min. The antigen signal was detected by incubating the slides in Chromogen IP Warp Red stain (BioCare Medical) for 10 min. Haematoxylin counterstaining was performed post-antigen staining.

## Software
Figures were generated using Python 3 v3.10[56] and the packages matplotlib v3.6.0[57], NumPy v1.23.3[58], pandas v1.5.0[59], and seaborn v0.12.0[60]. Simulations were conducted in Python 3 v3.10.

## Analysis of genotype frequencies, richness, and diversity
Here a viral genotype is defined as a unique combination of the eight IAV segments, where each segment is derived from either the variant or wild-type parental virus; therefore, there are $2^8$ possible unique genotypes, with two parental genotypes and 254 reassortant genotypes. For any given sample, the frequency of each unique genotype can be calculated by dividing the number of appearances each unique genotype has in the sample by the total number of clonal isolates obtained for that sample.

Understanding the distribution of unique genotypes involves using both unweighted and weighted genotype frequency statistics. Genotype richness ($S$) does not incorporate genotype frequency and is given by the number of unique genotypes in a sample. Given our sample size of 21 plaque isolates, genotype richness, or the number of distinct genotypes detected in a sample, can range from a minimum of 1 (a single genotype is detected 21 times) to a maximum of 21 (21 unique genotypes detected).

Diversity was measured using the Shannon–Weiner index ($H$), which considers both richness and evenness in the frequency with which genotypes are detected. In our dataset, diversity can range from 0 to 3.04. Shannon–Wiener diversity was calculated as:

$$H = - \sum_{i=1}^{S} (p_i * lnp_i) \tag{1}$$

where $S$ is genotype richness and $p_i$ is the frequency of unique genotype $i$ in the sample (6).

To address whether evaluating 21 plaques per sample for this analysis was sufficient to yield robust results on genotype diversity, we used a computational simulation to test the sensitivity of the measured diversity values to the number of plaques sampled. In these simulations, we calculated the diversity present in samples generated by randomly picking $n$ (out of the possible 21) plaques without replacement. At each sampling effort $n$, we simulated 1000 samples, with plaque replacement between samples. The results typically show that diversity values increase as $n$ increases, with values asymptoting as $n$ approaches 21, suggesting that further increases in $n$ would not greatly change results and validating the use of 21 plaques (Supplementary Fig. 10).

To evaluate the extent to which the spatial dynamics of viral reassortment and propagation shape the overall richness and diversity in a host, we sought to compare the observed richness and diversity at each anatomical site to that which would be expected if virus moves freely among anatomical locations. Thus, to simulate free mixing within the host, we randomly shuffled observed viral genotypes among all sites in a given animal. The average richness and Shannon–Wiener index of the simulated viral populations at each site were then calculated. The 5th and 95th percentiles for the simulated distribution of each animal were calculated and compared to the observed richness and diversity for each of the anatomical sites. If a site's observed richness and diversity fell below the 5th percentile or above the 95th percentile, then a barrier to the influx or efflux of reassortant genotypes from or to the other sites is suggested.

## Analysis of beta diversity
The dissimilarity between populations can be measured by beta diversity. For this study, we evaluated beta diversity from a richness perspective, focusing on dissimilarity in the unique genotypes detected and excluding consideration of their frequency. This approach was used to de-emphasize the effects of WT and VAR parental genotypes, which were likely seeded into all anatomical locations at the time of inoculation. We calculate the beta diversity by treating the viral genotypes in two lobes as two distinct populations:

$$\beta = \frac{S_{1+2}}{\frac{1}{2}(S_1 + S_2)} \tag{2}$$

where $S_{1+2}$ is the richness of a hypothetical population composed of pooling the viral genotypes of the two lobes while $\frac{1}{2}(S_1 + S_2)$ represents the mean richness of the lobes (7). The beta diversity of a single comparison can be normalized so that it ranges from zero to one:

$$BD' = \frac{BD - 1}{BD_{max} - 1} \beta_n \tag{3}$$

where $BD_{max}$ is the beta diversity calculated by assuming that there are no viral genotypes shared by both lobes (7). A $BD'/\beta_n$ closer to one indicates that the lobes' viral populations are more dissimilar while a $BD'\beta_n$ closer to zero suggests that the lobes have similar unique viral genotypes and overall viral richness. A $BD'\ \beta_n$ of zero occurs when all unique genotypes present in one lobe are also present in the other.

To address whether evaluating 21 plaques per sample for this analysis was sufficient to yield robust results on beta diversity, we again used computational simulations. These simulations were designed to test the sensitivity of beta diversity values to the number of plaques sampled. In these simulations, we again generated plaque data subsets by randomly picking $n$ (out of the possible 21) plaques without replacement. At each sampling effort $n$, we simulated 1000 samples, with plaque replacement between samples. Beta diversity values were then calculated based on these data subsets, at a given $n$. The results typically show that $\beta_n$ values tend to stabilize as $n$ approaches 21, suggesting that further increases in $n$ would not greatly

change results and validating the use of 21 plaques (Supplementary Fig. 10). In a subset of cases that involve the nasal sample of Pig 5, however, the relationship between $\beta_n$ and $n$ is less stable. In sharp contrast to most other samples from Pig 5, the nasal site showed 20 WT parental isolates and one VAR parental isolate. The lung tissues had no WT parental genotypes detected. As a result, each successive plaque draw from the nasal sample increases the probability of detecting the VAR virus and therefore detecting a commonality between the nasal tract and any of the lung lobes. Thus, in situations where two tissue sites have a single, relatively rare genotype in common, the number of plaques sampled has a strong impact on $\beta_n$ outcomes.

To simulate free mixing between two lobes, we randomly shuffled the genotypes between each of the 28 pairwise combinations among pig tissues and the single ferret lung-NT combination and computed the $BD'\beta_n$ for each comparison. Free mixing for all combinations was simulated 1000 times. We reasoned that if compartmentalization was present in the observed dataset, then the dissimilarity values would fall at the high end of the simulated distribution (>95th percentile).

### Statistical measures

Percentiles were calculated using the percentileofscore method from the SciPy package v1.9.1[59]. Paired $t$ tests and ANOVA tests were performed using the ttest_rel method and the f_oneway method respectively from the SciPy Package v1.9.1[59].

### Reporting summary

Further information on research design is available in the Nature Portfolio Reporting Summary linked to this article.

## Data availability

Raw data used for the generation of Figs. 1–5 and Supplementary Figures 1–10 are included as Source Data and in Supplementary Data 1–3. Source data are provided in this paper.

## Code availability

Code used for data analysis and simulations is available at: https://github.com/maxbagga/Influenza-A-virus-reassortment-in-mammals-gives-rise-to-spatially-distinct-sub-populations (https://doi.org/10.5281/zenodo.7150566)[61].

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

## Acknowledgements

We thank Shamika Danzy for their technical assistance. We would like to thank the Emory Division of Animal Resources for their assistance with the guinea pig experiments. We thank Rusty Ransburgh, Baoliang Zheng, Dongchang He, Abaineh Endalew, Daniel Madden, Yuekun Lang for their assistance with the pig experiments. We thank David VanInsberghe for assistance with figure generation. This project was supported by the Emory University Integrated Cellular Imaging Microscopy Core. The research was funded by NIH R01 AI127799 (A.C.L.) and the NIH/NIAID Centers of Excellence in Influenza Research and Response (CEIRR), contract numbers 75N93021C00017 (A.C.L., K.K.), 75N93021C00014 (D.R.P.) and 75N93021C00016 (J.A.R., W.M.), and the AMP Core of the Center of Emerging and Zoonotic Infectious Diseases (CEZID) from NIGMS under award number P20GM130448 (J.A.R.).

## Author contributions

K.G. contributed to the conception of work, experimental design, data acquisition and analysis, and interpretation of data; A.B. contributed to data analysis, model development, and interpretation of data; S.C. contributed to experimental design, data acquisition, and interpretation of data; L.M.F., G.G., C.J.C., B.S., Y.L., L.W., T.K., Y.L., and I.M. contributed to data acquisition; W.M., J.A.R., and D.R.P. contributed to the conception of the work, experimental design, data analysis, and interpretation; K.K. contributed to the conception of the work, data analysis and interpretation; A.C.L. contributed to the conception of work, experimental design, and data analysis and interpretation. All authors contributed to the writing of the manuscript.

## Competing interests

The JAR laboratory received support from Tonix Pharmaceuticals, Xing Technologies, and Zoetis, outside of the reported work. J.A.R. is inventor on patents and patent applications on the use of antivirals and vaccines for the treatment and prevention of virus infections, owned by Kansas State University, KS. The remaining authors declare no competing interests.

## Additional information

Anice C. Lowen.

**Peer review information** *Nature Communications* thanks Richard
Webby and the other, anonymous, reviewer(s) for their contribution to
the peer review of this work. Peer reviewer reports are available.

[1]Department of Microbiology and Immunology, Emory University School of Medicine, Atlanta, GA, USA. [2]Emory College of Arts and Sciences, Atlanta,
GA, USA. [3]Department of Population Health, College of Veterinary Medicine, University of Georgia, Athens, GA, USA. [4]Department of Diagnostic Medicine and
Pathobiology, College of Veterinary Medicine, Kansas State University, Manhattan, KS, USA. [5]Department of Veterinary Pathobiology, and Department of
Molecular Microbiology and Immunology, University of Missouri, Columbia, MO, USA. [6]St. Jude Center of Excellence for Influenza Research and Response
(SJ-CEIRR), Memphis, TN, USA. [7]The Center for Research on Influenza Pathogenesis and Transmission (CRIPT CEIRR), New York, NY, USA. [8]Department of
Biology, Emory University, Atlanta, GA, USA. [9]Emory Center of Excellence for Influenza Research and Response (Emory-CEIRR), Atlanta, GA, USA.
✉e-mail: anice.lowen@emory.edu

