## [Peer Review File · Nature Communications]

Influenza A virus reassortment in mammals gives rise to genetically distinct within-host sub-populationsReviewers' Comments:

Reviewer #1:

Remarks to the Author:

The manuscript by Ganti et al describes a comparative study of influenza virus reassortment dynamics in guinea pigs, ferrets, and swine. Using a mix of segment barcoded influenza, the authors find that reassortment mirrors the compartmentalisation of virus replication in different segments of the respiratory tract of each animal group and that influenza viruses appear to reassort less frequently in swine than in guinea pigs or ferrets.

The finding that influenza viruses reassort less freely in swine than in ferrets or guinea pigs is surprising, primarily because influenza viruses so frequently reassort in swine. One possible reason for this finding is the difference in experimental infection protocols in swine compared to the ferrets and guinea pigs. Swine were infected with larger doses and infected both intra-nasally and intra-tracheally compared to lower doses and only intranasal infection in both ferrets and guinea pigs. This difference in protocol could lead to greater and earlier saturation of susceptible host cells with the net effect of reducing the time for new reassortant populations to form and proliferate. Showing the virus titers from each animal over the course of the infection would help identify whether or not this is the case. Any other data to substantiate similar time courses of virus replication in each compartment of each animal group would be similarly helpful. It would seem important to highlight the swine protocol difference in the discussion and to elaborate on the potential consequences for the conclusion that reassortment is less frequent in swine. If the finding of less frequent reassortment in swine is real then it would help to further hypothesise in the discussion about why this might be.

The structure of the paper is somewhat peculiar to me with a substantial portion of the introduction devoted to the results and conclusions. Some restructuring of the paper to follow a more conventional flow would improve the manuscript.

Reviewer #2:

Remarks to the Author:

This manuscript represents a very nice set of experimental data concerning influenza virus reassortment. Reassortment is a central process in the emergence of novel viruses, as detailed by the authors, but is difficult to dissect and uncover fundamental mechanistic insights. As such, this manuscript provides interesting data that will be of broad interest. The authors are to be congratulated for including three different animal models in their work. Typing reassortant viruses, even with the smart methodology used in this study, is tedious. Specific comments follow.

1) As mentioned, this type of work is tedious and I completely understand the limitations involved.

But some discussion of what power selecting 21 viruses from each sample has would be of interest.

The authors cite a previous work in 433, but a discussion of the power of this in the context of the current study (even if a limitation which may or may not be the case) would be useful.

2) Unless I am misinterpreting the figures (which could be the case, I did find some of them a bit of a struggle to get through) there appears to be a disproportionate absence of the VAR parental genotype detected (for example Fig 1 A,E,I and Fig 4 A to F). This seems to be in conflict with the immunostaining which by eye seems to indicate more VAR (at least HA) infection. Do the authors have any thoughts on this? Why is VAR WT not detected frequently? Does it form the same size plaques as WT? The dominance of the WT parental genome suggests to me that some form of selection is occurring, even though the authors suggest this isn't the case.

3) Although I assume it is the case, were both viruses mixed together prior to inoculation of animals?

4) There isn't much of a discussion of the role of dose in the observed trends. Did dose have much of an impact on results? Were there any significant dose dependent trends?

5) Is it possible that less reassortment was seen in pigs simply because they are bigger animals with more respiratory cells and the effective MOI was far lower? This would lead to lower coinfections? Was

there any evidence from the immunohistochemistry of less coinfections (its a bit difficult to tell from the selected images)? This could also be extrapolated to why lungs had less reassortment than upper respiratory samples and one of the reasons behind the dose question above.

6) How were ferret lungs treated? were individual lobes pooled?

Reviewer #3:

Remarks to the Author:

The study by Ganti et al. addresses a long-standing question in the field that was not systematically studied yet. Analyzing reassortment events in vivo and particularly in the respiratory tract will give important insights into viral diversity and evolution processes. Although this study makes a promising start, the data are unfortunately very preliminary at this stage to draw firm conclusions.

Major concerns:

#1: Figure 1: The dose used for the inoculation of pigs is too high. This can be seen in Figure 1J, where 3 of 6 animals do not show productive replication as one would expect in a highly susceptible host. In these 3 of 6 animals, virus titers drop at day 3 p.i. unlike in the low-dose infected ferrets (albeit some high-dose infected ferrets show the same picture). This is likely due to the high virus dose used. It is known in evolution biology that high doses do not favor virus evolution and thus diversity. Therefore, productive replication over a few time points is absolutely essential to study virus evolution and diversity thoroughly.

#2: The observation in ferrets and guinea pigs (where also naturally virus replication is limited to 1-2 time points, see Figure 1B) is nevertheless solid and interesting suggesting higher diversity in the URT. However, important questions were not addressed, such as: which RT cell subsets drive virus diversity? type I or II pneumocytes? These could have been addressed using single-cell sequencing technology. URT diversity observed would suggest increasing the likelihood for the generation of more transmissible virus strains. Transmission experiments would have been very interesting and more insightful regarding the meaning of the events.

#3: from the evolution perspective, including bird species, such as ducks or chicken would have clearly strengthened the study.

Reviewer #1 (Remarks to the Author):

The manuscript by Ganti et al describes a comparative study of influenza virus reassortment dynamics in guinea pigs, ferrets, and swine. Using a mix of segment barcoded influenza, the authors find that reassortment mirrors the compartmentalization of virus replication in different segments of the respiratory tract of each animal group and that influenza viruses appear to reassort less frequently in swine than in guinea pigs or ferrets.

The finding that influenza viruses reassort less freely in swine than in ferrets or guinea pigs is surprising, primarily because influenza viruses so frequently reassort in swine. One possible reason for this finding is the difference in experimental infection protocols in swine compared to the ferrets and guinea pigs. Swine were infected with larger doses and infected both intranasally and intra-tracheally compared to lower doses and only intranasal infection in both ferrets and guinea pigs. This difference in protocol could lead to greater and earlier saturation of susceptible host cells with the net effect of reducing the time for new reassortant populations to form and proliferate. Showing the virus titers from each animal over the course of the infection would help identify whether or not this is the case. Any other data to substantiate similar time courses of virus replication in each compartment of each animal group would be similarly helpful. It would seem important to highlight the swine protocol difference in the discussion and to elaborate on the potential consequences for the conclusion that reassortment is less frequent in swine. If the finding of less frequent reassortment in swine is real, then it would help to further hypothesize in the discussion about why this might be.

We thank the reviewer for highlighting this issue. We have added data on the dynamics of viral loads in the nasal tract for all infected animals as Supplementary Figure 2. We have also taken care to highlight the protocol difference in pigs relative to ferrets and guinea pigs in the results at lines 86-93 and discuss the potential implications of this difference at lines 118-122. Finally, we now include text in the discussion beginning at line 233 which addresses the potential implications of this protocol difference and possible reasons why observed reassortment may be lower in pigs than in the other species. This text is reproduced here:

“In drawing comparisons across the species examined, it is important to consider differences in the protocols applied. Swine were infected both intranasally and intratracheally compared to only intranasal infection in both ferrets and guinea pigs. The delivery of inoculum directly to the lower respiratory tract in swine might be expected to augment potential for wt-var coinfection in the lungs. In addition, swine were infected with a single high dose with unknown relationship to the median infectious doses in this species, while ferrets and guinea pigs were each infected with doses of 1×10^2 and 1×10^5 ID₅₀. As measured by plaque assay, the dose applied in pigs is comparable to the high dose used in guinea pigs and approximately 10-fold lower than the high dose used in ferrets. While inoculum dose might be expected to influence the potential for coinfection and propagation of novel variants, the effect of a 10³-fold difference in dose in ferrets and guinea pigs was modest. We therefore suggest that the more limited diversification through reassortment in pigs is unlikely to be due to protocol differences.

Given our use of well-matched parental viruses, the differences in viral genotypic diversity seen between species and between tissues within a species are also

unlikely to be driven by selection. Instead, features of viral dynamics in these different environments likely shape the frequency of reassortment and the extent to which reassortants are amplified once formed. Since relatively few reassortants were detected in pigs and in ferret lungs, our data imply that the size of the target tissue may be an important factor. In a large animal such as swine or a tissue with extensive surface area, such as lung, the likelihood of cellular co-infection involving genetically distinct variants may be reduced. In addition, viral fitness, the spatial arrangement of target cells, the extent to which permissive cells are interspersed with non-permissive cells, physical barriers to viral spread and the effectiveness innate responses to infection are all likely to vary with species and across different tissues within a species³⁴⁻³⁶.”

The structure of the paper is somewhat peculiar to me with a substantial portion of the introduction devoted to the results and conclusions. Some restructuring of the paper to follow a more conventional flow would improve the manuscript.

This structure came about as we sought to limit the length of the Introduction section to meet journal requirements. We have made some structural changes and added mention of recently published work in the introduction.

Reviewer #2 (Remarks to the Author):

This manuscript represents a very nice set of experimental data concerning influenza virus reassortment. Reassortment is a central process in the emergence of novel viruses, as detailed by the authors, but is difficult to dissect and uncover fundamental mechanistic insights. As such, this manuscript provides interesting data that will be of broad interest. The authors are to be congratulated for including three different animal models in their work. Typing reassortant viruses, even with the smart methodology used in this study, is tedious. Specific comments follow.

1) As mentioned, this type of work is tedious, and I completely understand the limitations involved. But some discussion of what power selecting 21 viruses from each sample has would be of interest. The authors cite a previous work in 433, but a discussion of the power of this in the context of the current study (even if a limitation which may or may not be the case) would be useful.

We thank the reviewer for raising this point. To address whether evaluating 21 plaques per sample for this analysis was sufficient to yield robust results, we used a computational simulation to test the sensitivity of the measured diversity and beta diversity values to plaque number. In these simulations, we generated plaque data subsets by randomly picking n (out of the possible 21) plaques without replacement. At each sampling effort n , we simulated 1000 samples, with plaque replacement between samples. Diversity or beta diversity values were then calculated based on these data subsets, at a given n . The results of this analysis are included as Supplementary Figure 10.

The results typically show that diversity or beta diversity values stabilize as n approaches 21, suggesting that further increases in n would not greatly change results. In a subset of cases that involve the nasal sample of Pig 5, however, the relationship

between beta diversity and n is less stable. In sharp contrast to most other samples from Fig 5, the nasal site showed 20 WT parental isolates and one VAR parental isolate. The lung tissues had no WT parental genotypes detected. As a result, each successive plaque draw from the nasal sample increases the probability of detecting the VAR virus and therefore detecting a commonality between the nasal tract and any of the lung lobes. This effect is potent because the form of beta diversity that we have applied counts the detection of a given genotype at any frequency as the same. Thus, in situations where two tissue sites have a single, relatively rare, genotype in common, the number of plaques sampled has a strong impact on beta diversity outcomes. We have discussed this limitation in our approach at lines 304-308 (reproduced here):

“In addition, if variants are rare (<5%), they would fall below the limit of detection of our assay, which can have potent effects on measured beta diversity. In particular, if a genotype detected in one location is present below the limit of detection in a second location, measured beta diversity would not reflect this commonality and would therefore be erroneously high.”

2) Unless I am misinterpreting the figures (which could be the case, I did find some of them a bit of a struggle to get through) there appears to be a disproportionate absence of the VAR parental genotype detected (for example Fig 1 A,E,I and Fig 4 A to F). This seems to be in conflict with the immunostaining which by eye seems to indicate more VAR (at least HA) infection. Do the authors have any thoughts on this? Why is VAR WT not detected frequently? Does it form the same size plaques as WT? The dominance of the WT parental genome suggests to me that some form of selection is occurring, even though the authors suggest this isn't the case.

In this case, the reviewer has modestly mis-interpreted the figures. The predominant viral genotype detected in all animals is the VAR virus (shown in blue in the pie charts). Correspondingly, the predominant viral antigen detected is the VAR HA (shown in red in the tissue sections).

The reviewer is, however, correct that the two parental viruses are not detected with equivalent frequencies, despite being combined 1:1 in the inoculum. The same virus mixture was used to inoculate all animals evaluated here, so this skewing could be due to a deviation from 1:1 in the inoculum. Alternatively, fitness differences may be at play. This is now mentioned at lines 107-109. In addition, to evaluate the second possibility, we now include a multicycle growth analysis of NL09-WT and NL09-VAR viruses in cell culture as Supplementary Figure 1 (and find that viral replication in this context is indistinguishable).

3) Although I assume it is the case, were both viruses mixed together prior to inoculation of animals?

Yes. This has been clarified in the Methods section at lines 356-359.

4) There isn't much of a discussion of the role of dose in the observed trends. Did dose have much of an impact on results? Were there any significant dose dependent trends?

We thank the reviewer for raising this point. Minimal discussion of dose was included because we saw little effect of dose, but we now include a formal analysis included as Supplementary Figure 4 and discussed at lines 113-117 (reproduced below):

“In guinea pigs and ferrets, only marginal effects of dose were apparent (Fig 1, compare dashed and solid lines). Nevertheless, differences in diversity were significant in both species, with the higher dose yielding higher diversity when all time points were considered together ($p=0.04$ in guinea pigs and $p=0.003$ in ferrets, ANOVA). Richness was also significantly higher in ferrets receiving a higher dose ($p=0.002$, ANOVA).“

5) Is it possible that less reassortment was seen in pigs simply because they are bigger animals with more respiratory cells and the effective MOI was far lower? This would lead to lower coinfections? Was there any evidence from the immunohistochemistry of less coinfections (its a bit difficult to tell from the selected images)? This could also be extrapolated to why lungs had less reassortment than upper respiratory samples and one of the reasons behind the dose question above.

We agree with the reviewer’s line of thinking and have added discussion of the possible role of tissue/animal size in determining the likelihood of coinfection. The new text is at lines 248-252 and is reproduced here:

“Since relatively few reassortants were detected in pigs and in ferret lungs, our data imply that the size of the target tissue may be an important factor. In a large animal such as swine or a tissue with extensive surface area, such as lung, the likelihood of cellular co-infection involving genetically distinct variants may be reduced.”

6) How were ferret lungs treated? were individual lobes pooled?

A single lobe was sampled from ferrets. This has been clarified in the Methods section at line 408-409.

Reviewer #3 (Remarks to the Author):

The study by Ganti et al. addresses a long-standing question in the field that was not systematically studied yet. Analyzing reassortment events in vivo and particularly in the respiratory tract will give important insights into viral diversity and evolution processes. Although this study makes a promising start, the data are unfortunately very preliminary at this stage to draw firm conclusions.

We respectfully disagree that the study is preliminary given the use of three different animals models with both temporal and spatial sampling and in-depth analysis of the resultant data using multiple quantitative and computational approaches.

Major concerns:

#1: Figure 1: The dose used for the inoculation of pigs is too high. This can be seen in Figure 1J, where 3 of 6 animals do not show productive replication as one would expect in a highly susceptible host. In these 3 of 6 animals, virus titers drop at day 3 p.i. unlike in the low-dose infected ferrets (albeit some high-dose infected ferrets show the same picture). This is likely due to the high virus dose used. It is known in evolution biology that high doses do not favor virus evolution and thus diversity. Therefore, productive replication over a few time points is absolutely essential to study virus evolution and diversity thoroughly.

We note that Figure 1 does not show viral loads. The panel referred to shows the frequency of parental viral genotypes in swine nasal samples. That these frequencies decline early indicates that the frequency of reassortant genotypes increased between the first and second time points. We remind the reviewer that reassortment is not possible without productive replication. To clarify this point, we have now included viral load data as Supplementary Figure 2 in the revised manuscript.

The dose used in pigs was 2×10^6 PFU per animal. Dosing within this order of magnitude is standard for swine models and was selected in our study following a failed attempt to use lower doses. Lower doses did not consistently lead to productive infection. This is now noted in the results section at lines 89-91.

#2: The observation in ferrets and guinea pigs (where also naturally virus replication is limited to 1-2 time points, see Figure 1B) is nevertheless solid and interesting suggesting higher diversity in the URT. However, important questions were not addressed, such as: which RT cell subsets drive virus diversity? type I or II pneumocytes? These could have been addressed using single-cell sequencing technology. URT diversity observed would suggest increasing the likelihood for the generation of more transmissible virus strains. Transmission experiments would have been very interesting and more insightful regarding the meaning of the events.

It is also relevant to this comment that Figure 1 does not show viral loads. Contrary to the reviewer's interpretation, guinea pigs and ferrets shed virus for 6-7 days post-inoculation. These data are included as Supplementary Figure 2 in the revised manuscript.

We agree that there may be interesting differences among cell types within the respiratory tract in terms of the likelihood of coinfection and the role in generating reassortant viral diversity. To evaluate the cell types infected in ferret nasal turbinate, ferret lung and pig lung samples, we now include additional immunohistochemistry analyses designed to enable morphological identification of cell types by a board certified veterinary pathologist. These results are included in the text at lines 172-176 and 216-220, and are shown in Figures 3 and 5 for ferrets and pigs, respectively. In both pigs and ferrets, ciliated cells within the bronchi and bronchioles and type II pneumocytes within the alveoli were major targets. Additional cell types were infected in ferrets including goblet cells and submucosal glands. Unfortunately, appropriate tissue samples to evaluate ferret nasal turbinate could not be identified (relevant regions of the tissue blocks were exhausted) but in pigs respiratory and transitional epithelia were found to be positive at the nasal site. Of note, the cell types infected in the lung are not present in the nasal tract, so a difference of cell types between upper and lower respiratory tracts is a possible driver of differences in diversity seen in ferrets.

Regarding transmission, we feel that this topic is outside of the scope of the current manuscript, which is focused on within-host viral dynamics. We and others have furthermore previously examined the anatomical site at which IAV transmission initiates and found clear evidence that the upper respiratory tract is the source of transmitted virus (e.g. Richard et al. 2020). This point is already discussed in the Discussion section. In addition, in the guinea pig model, we previously showed that diverse reassortant viruses generated during the course of infection in a donor animal are transmitted onward to naïve contacts (Tao et al. 2015).

#3: from the evolution perspective, including bird species, such as ducks or chicken would have clearly strengthened the study.

We have evaluated reassortment in mallards recently and this work is cited in the manuscript (Ganti et al. 2021). We note, however, that it is difficult to compare reassortment rates between avian and mammalian species since the same IAV strains are not relevant in each. For the mallard work, we used mallard-derived viruses that would replicate poorly in mammals. We have furthermore found over the last several years that differing IAV strains have intrinsically different reassortment rates, depending on the extent to which coinfection augments viral productivity (Phipps et al. 2020). Assigning differences in reassortment rates to host species rather than virus strains is therefore problematic.

Reviewers' Comments:

Reviewer #1:

Remarks to the Author:

My concerns have been addressed.

Reviewer #2:

Remarks to the Author:

The authors have responded well to my suggestions (and those of the other reviewers) leading to an improved manuscript. I have no further comments.

Reviewer #3:

Remarks to the Author:

Unfortunately, my major concerns #1 and #2 have not been thoroughly addressed. The authors at least now discuss the major limitation of the study regarding the differences in the infection dosing used (as also highlighted by reviewer #1). However, this still does not allow any firm conclusions regarding virus evolution. It is evident from Figure 1 and also from the new Supplementary Figure 2 that productive virus replication does not occur in pigs (maybe with the exception of P4 and P6, see Figure 1). This then, regrettably, does not allow a solid conclusion on intra and inter virus evolution and with that the working hypothesis remains still open.

Reviewer #1 (Remarks to the Author):

My concerns have been addressed.

We thank the reviewer for their comments

Reviewer #2 (Remarks to the Author):

The authors have responded well to my suggestions (and those of the other reviewers) leading to an improved manuscript. i have no further comments.

We thank the reviewer for their comments

Reviewer #3 (Remarks to the Author):

Unfortunately, my major concerns#1 and #2 have not been thoroughly addressed. The authors at least now discuss the major limitation of the study regarding the differences in the infection dosing used (as also highlighted by reviewer#1). However, this still does not allow any firm conclusions regarding virus evolution. It is evident from Figure 1 and also from the new Supplementary Figure 2 that productive virus replication does not occur in pigs (maybe with the exception of P4 and P6, see Figure 1). This then, regrettably, does not allow solid conclusion on intra and inter virus evolution and with that the working hypothesis remains still open.

Unfortunately, this reviewer is still misinterpreting the data in question here, which does not show viral load, but instead shows genotypic frequencies. As we have clearly shown in the new Supplementary Figure 2, all animals tested have productive viral replication as evidenced by plaque assay titers. We, therefore, respectfully disagree with this reviewer's comments.